# High-fidelity entanglement between a trapped ion and a telecom photon via quantum frequency conversion

Matthias Bock[1], Pascal Eich[1], Stephan Kucera[1], Matthias Kreis[1], Andreas Lenhard [1], Christoph Becher [1] & Jürgen Eschner [1]

Entanglement between a stationary quantum system and a flying qubit is an essential ingredient of a quantum-repeater network. It has been demonstrated for trapped ions, trapped atoms, color centers in diamond, or quantum dots. These systems have transition wavelengths in the blue, red or near-infrared spectral regions, whereas long-range fiber-communication requires wavelengths in the low-loss, low-dispersion telecom regime. A proven tool to interconnect flying qubits at visible/NIR wavelengths to the telecom bands is quantum frequency conversion. Here we use an efficient polarization-preserving frequency converter connecting 854 nm to the telecom O-band at 1310 nm to demonstrate entanglement between a trapped $^{40}Ca^+$ ion and the polarization state of a telecom photon with a high fidelity of 98.2 ± 0.2%. The unique combination of 99.75 ± 0.18% process fidelity in the polarization-state conversion, 26.5% external frequency conversion efficiency and only 11.4 photons/s conversion-induced unconditional background makes the converter a powerful ion–telecom quantum interface.

[1] Fachrichtung Physik, Universität des Saarlandes, Campus E2.6, 666123 Saarbrücken, Germany. These authors contributed equally: Matthias Bock, Pascal Eich. Correspondence and requests for materials should be addressed to C.B. (email: christoph.becher@physik.uni-saarland.de) or to J.E. (email: juergen.eschner@physik.uni-saarland.de)

Quantum repeaters that establish long-distance entanglement are essential tools in the emerging field of quantum communication technologies[1]. While proposals for memoryless repeaters exist (e.g., ref. [2]), many currently pursued approaches require efficient, low-noise quantum memories as nodes that exchange quantum information via photonic channels[3]. Various atomic and solid-state systems have been identified as suitable quantum nodes, e.g., trapped ions[4,5], trapped atoms[6,7], color centers in diamond[8], or quantum dots[9,10]. Their optical transitions, however, are—with few exceptions—located outside the wavelength regime between 1260 and 1625 nm, where telecom fibers afford low-loss transmission. Thus, there is a demand for interfaces connecting the telecom-wavelength regime and the visible/NIR range in a coherent way, i.e., preserving quantum information encoded in a degree of freedom of a single photon, such as its polarization.

Promising candidates for such interfaces are, e.g., non-degenerate photon-pair sources[11–13] or quantum frequency converters (QFC)[14]. The latter can be implemented either by four-wave mixing (FWM) using resonances in cold atomic ensembles[15,16] or by a solid-state approach utilizing three-wave mixing in $\chi^2$- or FWM in $\chi^3$-nonlinear media[17]. It has been shown that $\chi^2$-based QFC preserves nonclassical properties of single photons and photon pairs, such as second-order coherence[18–20], time-energy entanglement[21], time-bin entanglement[22], orbital angular momentum entanglement[23], polarization entanglement[24], and photon indistinguishability[25,26]; furthermore, nonclassical correlations between telecom photons and spin waves in cold atomic ensembles[27,28] have been demonstrated. Using near-resonant QFC based on FWM in an atomic ensemble, entanglement of a spin–wave qubit with the polarization state of a telecom photon has been realized[16]. A corresponding

implementation using solid-state QFC has remained an open challenge, despite being a highly desirable approach for its wavelength flexibility: while atomic ensembles are restricted to the particular transition wavelengths of neutral atoms, solid-state QFC can be adjusted to the system wavelength of other promising stationary quantum bits for quantum nodes, such as trapped ions, color centers in diamond or rare-earth ensembles. The main obstacle has been the strong polarization dependence of the $\chi^2$-process and the high demands on efficiency and noise properties of the converter. Despite successful attempts to overcome the polarization dependency[24,29,30], the integration of a solid-state QFC device that fulfills all above mentioned requirements into a quantum node has not been achieved.

Single trapped ions are promising systems for quantum nodes, providing a very high level of control over their photonic interaction[5,31,32] and large coherence times[33,34]; importantly, single-ion qubits are directly addressable and thus allow quantum information processing via high-fidelity quantum gates[35,36]. In our work, we connect a trapped-ion quantum node via QFC to the telecom regime in a coherent way, creating high-quality entanglement between the ion and a telecom photon. To this end, we generate entanglement between an atomic quantum bit in a single trapped $^{40}Ca^+$ ion and the polarization state of a single photon at 854 nm. Subsequent polarization-preserving QFC to the telecom O-band establishes high-fidelity entanglement between ion and telecom photon, which we verify by quantum-state tomography.

## Results

**Ion–photon quantum interface.** The ion–photon interface is shown in Fig. 1. A single $^{40}Ca^+$ ion is confined in a linear Paul trap and laser-excited. Photons at 854 nm emitted along the

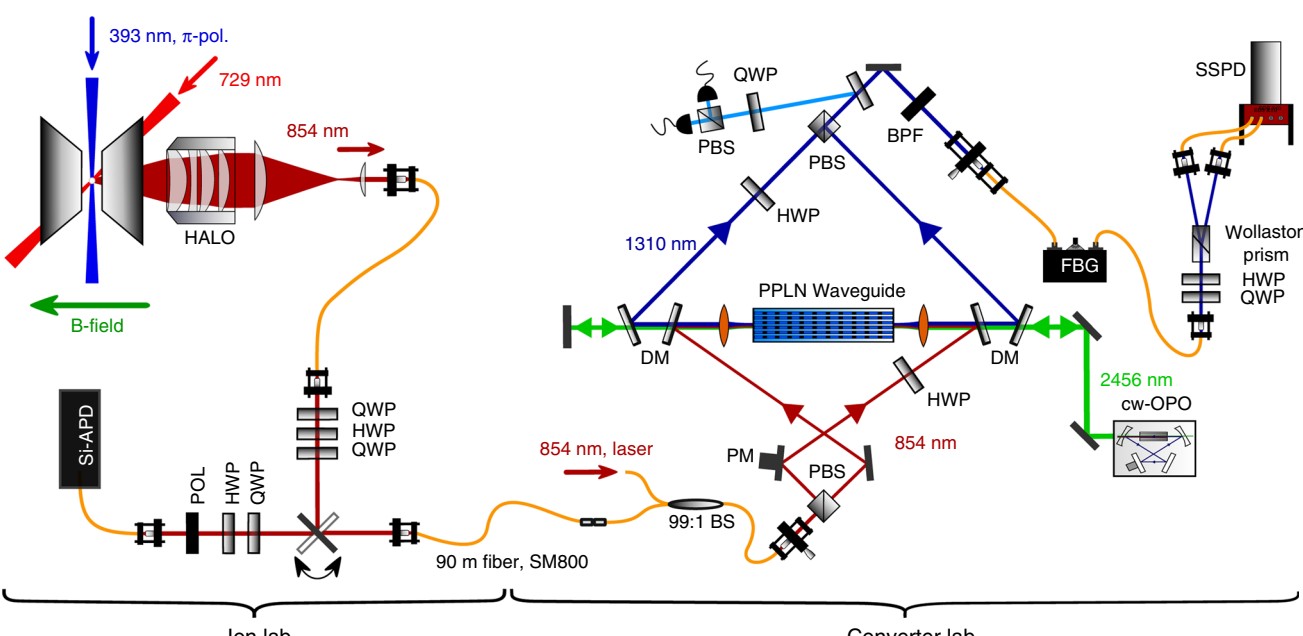

**Fig. 1** Experimental setup. Atom–photon entanglement is generated between a single trapped $^{40}Ca^+$-ion, confined and laser-cooled in a linear Paul trap, and a single photon at 854 nm. The photons are collected with a HALO ("High-numerical-Aperture Laser Objective") and coupled to a single-mode fiber. A combination of two quarter waveplates (QWP) and one half wave plate (HWP) is inserted behind the fiber to compensate all unitary rotations of the polarization state caused by the single-mode fiber and several dichroic mirrors between the fiber and the HALO. A flip mirror can be inserted behind the fiber to send the photons either to the converter lab via a 90 m long single-mode fiber or to the projection setup for 854 nm. The polarization-preserving frequency converter is realized with a nonlinear waveguide crystal in a single-crystal Mach–Zehnder configuration (for details see main text and Methods). The converted photons are detected with superconducting single-photon detectors. PBS polarizing beam splitter, DM dichroic mirror, BS beam splitter, BPF band pass filter, FBG fiber Bragg grating

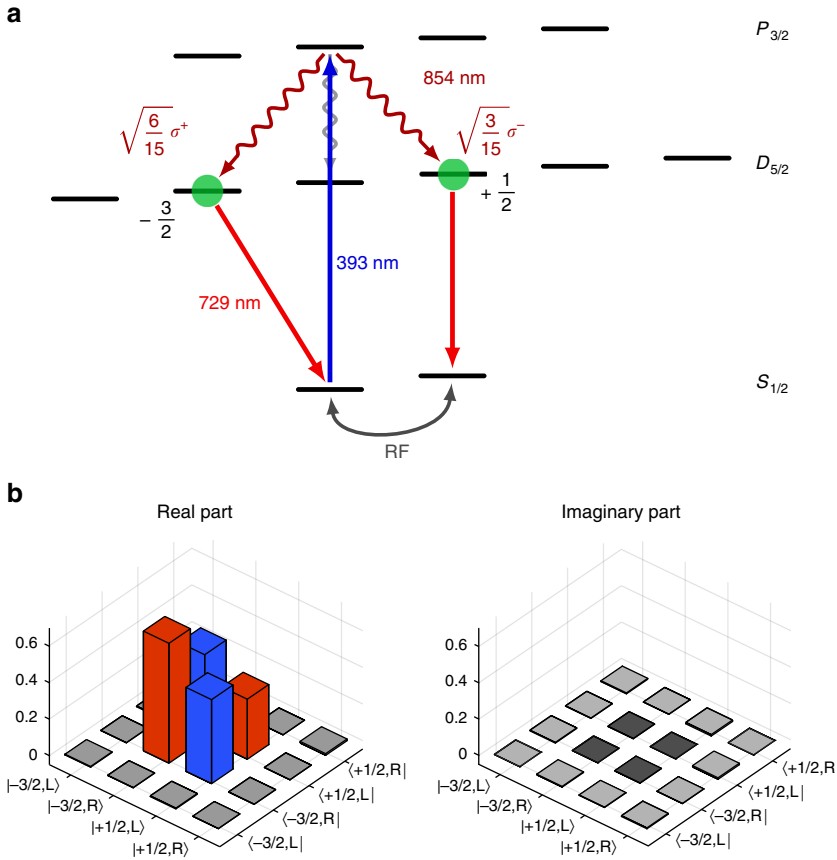

**Fig. 2** Ion–photon entanglement scheme and quantum-state tomography. **a** Atom–photon entanglement is generated via spontaneous decay from the $P_{3/2}$-to the $D_{5/2}$-state after excitation with $\pi$-polarized laser light at 393 nm. Emitted photons at 854 nm are collected along the quantization axis, thereby suppressing $\pi$-polarized photons. Atomic state analysis is realized via coherent pulses on the optical transition at 729 nm and the RF-transition in the ground state followed by fluorescence detection (details see Supplementary Note 1). **b** Real and imaginary part of the density matrix of the ion–photon entangled state, measured via quantum-state tomography. The different heights of the diagonal elements (red bars) results from the different Clebsch–Gordan coefficients of the $\sigma^{+}$- and the $\sigma^{-}$-transitions

quantization axis—defined by a magnetic field—are collected with 3.6% efficiency by a HALO ("High-numerical-Aperture Laser Objective", NA = 0.4) and coupled to a single-mode fiber with 39% efficiency (for details see Methods section). Further details on the setup are found in earlier publications[31,32,37]. The experimental sequence to generate atom–photon entanglement is shown in Fig. 2a. Starting from the ground state $S_{1/2}$, the ion is excited to the short-lived $P_{3/2}$ state with $\pi$-polarized laser light at 393 nm. Spontaneous decay to $D_{5/2}$ leads to entanglement between the atomic states $|-3/2\rangle = \left|D_{5/2}, m = -3/2\right\rangle$ and $|1/2\rangle = \left|D_{5/2}, m = 1/2\right\rangle$ and the emitted 854 nm photon in the polarization states $|R\rangle$ and $|L\rangle$. For details on the sequence see Supplementary Note 1. Taking into account the Clebsch–Gordan coefficients (CGC) of the two transitions (see Fig. 2a), the ideal ion–photon state is

$$|\Psi_{ideal}\rangle = \sqrt{\frac{2}{3}}|R, -3/2\rangle + \sqrt{\frac{1}{3}}|L, 1/2\rangle. \quad (1)$$

The experimentally generated ion–photon state is characterized by quantum-state tomography (see Methods). The real and imaginary parts of the reconstructed density matrix $\rho$ are shown in Fig. 2b. From $\rho$, we deduce the fidelity $F = \langle\Psi_{ideal}|\rho|\Psi_{ideal}\rangle$, denoting the overlap between the generated and the ideal state,

and the purity $P = \text{Tr}(\rho^2)$, a measure for the depolarization of the state. We find $F = 98.3 \pm 0.3\%$ and $P = 96.7 \pm 1.6\%$. An upper bound of the fidelity for a given purity is $F_{max} = \frac{1}{2}(1 + \sqrt{2P - 1})$ = 98.3% indicating that the fidelity is solely limited by depolarization and not by undesired unitary rotations of the state[11]. Depolarization is mainly caused by polarization-dependent loss in the optics behind the ion trap; minor contributions arise from non-perfect readout pulses and loss of atomic coherence. Calculating the overlap with a maximally entangled Bell-state yields $F_{Bell} = 95.5 \pm 0.3\%$. The maximum possible value of $F_{Bell}$ for our state is 97% (for $P = 1$), due to the asymmetric CGCs. Note that these numbers are calculated after subtraction of the detector dark counts, to characterize the functionality of our method. In a realistic repeater scenario these dark counts have to be included. Without background subtraction we deduce a purity $P = 92.1 \pm 1.6\%$ and fidelities $F = 95.9 \pm 0.3\%$ and $F_{Bell} = 93.3 \pm 0.3\%$. Even without background subtraction, all fidelities are many standard deviations above the classical threshold of 50%, as well as above the threshold of 70.7% necessary to violate Bell's inequalities. The method to perform the background subtraction along with tables summarizing all fidelities and purities can be found in Supplementary Note 7 and Supplementary Tables 1, 2 and 3. With a sequence repetition rate of ~58 kHz, we obtain 236 generated (27.6 projected and detected) entanglement events/s, which compares well with other

**a**

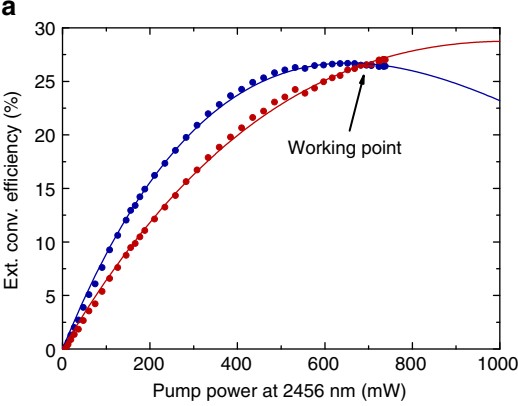

**b**

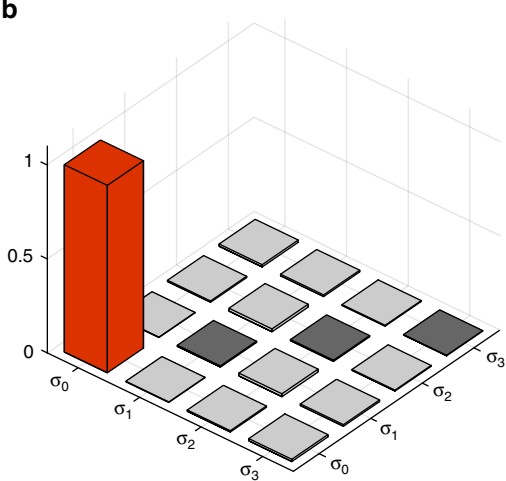

**Fig. 3** Characterization of the quantum frequency converter. **a** The external conversion efficiency of the two interferometer arms depending on the pump power of the mixing field at 2456 nm. The data points are fitted with $\eta_{\text{ext}}(P_{\text{P}}) = \eta_{\max}\sin^2\left(\sqrt{\eta_{\text{nor}}P_{\text{P}}}L\right)$[39]. **b** The absolute values of the process matrix of the coherent polarization-preserving down-conversion measured with a laser. The process fidelity is determined by the element corresponding to the identity operation (red bar). We achieve a value of 99.75 ± 0.18%. The error bars are deduced from a Poissonian distribution

Ca$^+$-ion systems[5]. One order of magnitude higher entanglement rates were reported in a Yb$^+$-system, mainly due to a higher sequence repetition rate enabled by shorter cooling times and the use of ultrafast laser pulses[38]. Our signal-to-background ratio (SBR) is 29.5, solely limited by detector dark counts. A detailed account of the derivation of all numbers is given in Supplementary Notes 4 and 5.

**Polarization-preserving quantum frequency converter**. The ion-trap setup is connected by 90 m of fiber to the converter setup, where the ion-entangled photons at 854 nm are converted to 1310 nm employing a periodically poled lithium niobate (PPLN) ridge waveguide designed for the difference-frequency mixing (DFG) process $1/_{854\,\text{nm}} - 1/_{2456\,\text{nm}} = 1/_{1310\,\text{nm}}$. As shown in Fig. 1, the polarization selectivity of the DFG process is overcome using a polarization interferometer: an arbitrary input state is split into H- and V-polarizations, a HWP rotates the H-polarization to the convertible V-polarization, and both are coupled via dichroic mirrors and zinc selenide aspheric lenses to the same waveguide from opposite directions. The V-polarized strong pump field at 2456 nm, generated by a home-built optical

parametric oscillator (OPO, see Methods) is aligned in double-pass configuration to facilitate conversion in both directions. The two converted polarizations are separated with dichroic mirrors from the other beams and superimposed on a second polarizing beam splitter (PBS) after undoing the rotation from H to V with another HWP. To ensure faithful conversion of arbitrary input polarizations, the path length of the interferometer is actively stabilized by injecting light from a stabilized diode laser at 854 nm via a chopper wheel and a 99:1 beam splitter into the setup. A second 99:1 BS splits a part of the converted laser light from the photon path, with which we measure the path length difference (light blue beam path). Feedback on the path length is realized by a piezo actuator connected to one of the mirrors (PM). Spectral filtering of the converted light with a broadband band pass filter (25 nm) and a narrowband fiber Bragg grating (25 GHz) suppresses the remaining pump light as well as noise arising from non-phase-matched nonlinear processes. With these two filter stages, the conversion-induced unconditional noise is reduced to 11.4 photons/s. The external conversion efficiency $\eta_{\text{ext}}$ (defined as "fiber-to-fiber" efficiency of the complete QFC device) of the two interferometer arms vs. the power of the pump field at 2456 nm, $P_{\text{P}}$, is shown in Fig. 3a. The data points are described quite well by the theoretical curve $\eta_{\text{ext}}(P_{\text{P}}) = \eta_{\text{ext,max}}\sin^2\left(\sqrt{\eta_{\text{nor}}P_{\text{P}}}L\right)$[39]. However, the setup is not fully symmetric with respect to forward and backward conversion: due to losses in the optics behind the waveguide, the backwards-propagating pump power is lower, thus the curve of the V-polarized arm (red data points) is shifted to higher pump powers. Nevertheless, we identify a working point at the intersection of the curves, which ensures an equal conversion efficiency of 26.5% for H- and V-polarized light, which compares well with other QFC systems[18,20,22,25,30]. To verify that the converter preserves arbitrary input polarization states, we apply process tomography[40] using laser photons. We prepare four different input states {H, V, D, L} and measure the respective Stokes vectors of the output state. With that the process matrix $\chi$ in the Pauli basis, connecting the in- and output density matrices via $\rho_{\text{out}} = \sum_{mn}\chi_{mn}\sigma_m\rho_{\text{in}}\sigma_n^\dagger$, is calculated (Fig. 3b). In the ideal case, $\chi$ possesses only a single non-zero entry $\chi_{00}$ denoting the identity operation. This entry can be identified as process fidelity, which in our case is $F_{\text{pro}} = 99.75 \pm 0.18\%$, confirming very high-fidelity conversion of the input polarization state. The error in $F_{\text{pro}}$ is deduced from a Poissonian distribution and arises from power fluctuations of the input and the pump field. Further details on the converter are given in the method section.

**Ion–telecom–photon entanglement**. To characterize the full quantum interface we investigate the performance of the combined ion-converter system: detecting the telecom output photons on a superconducting single-photon detector (SSPD) yields 43.5 generated (24.8 projected/detected) events/s with a SBR of 24.3. These numbers are in very good agreement with the previously determined conversion and detection efficiencies (see Supplementary Notes 4 and 5). Despite the loss in the conversion, the SBR is only weakly affected as we benefit from the detector's higher efficiency and lower dark-count rate. The density matrix of the ion–photon state after conversion is depicted in Fig. 4a, yielding $F = 97.7 \pm 0.2\%$, $P = 95.8 \pm 1.3\%$, and $F_{\text{Bell}} = 94.8 \pm 0.2\%$ after background subtraction. This result unambiguously verifies the entanglement between the ion and the telecom photon after QFC. The reduction of the fidelity by 0.6% compared to the unconverted ion–photon state is higher than what we expect from the process fidelity. We attribute this to power fluctuations of the pump laser and slow polarization drifts in the fiber connecting the setups. The background in these measurements has two contributions: a minor part of 6.5% due to conversion-induced noise and a major part of 93.5%

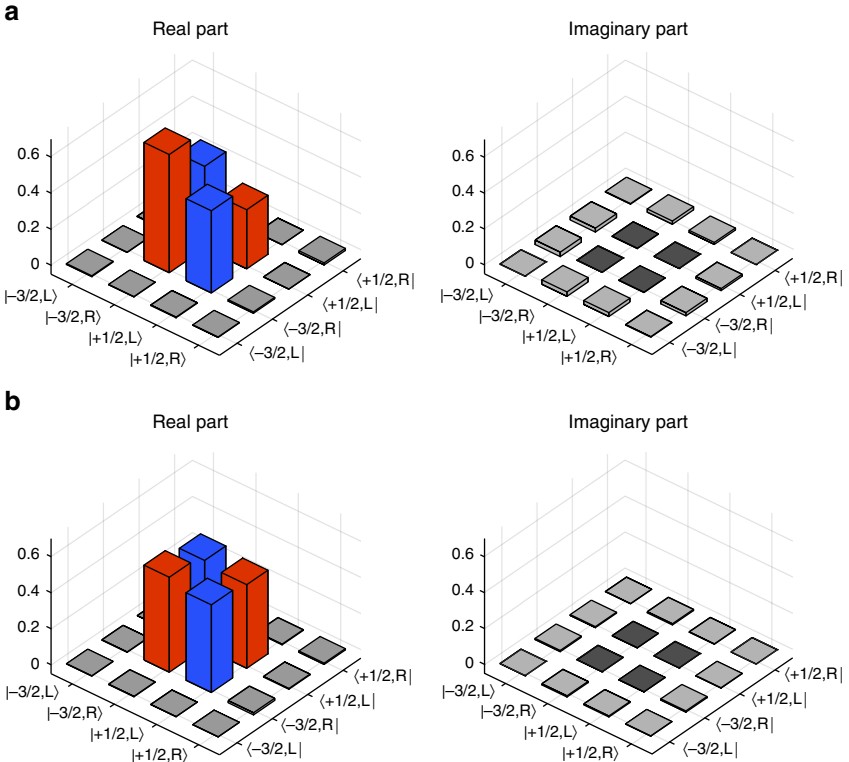

**Fig. 4** Quantum-state tomography of the ion–telecom-photon entangled states. Real and imaginary parts of the density matrices for: **a** the converted "bare" ion–photon entangled state still revealing the asymmetry in the diagonal elements (red bars) due to the asymmetric Clebsch–Gordan coefficients, **b** the converted ion–photon entangled state projected onto a Bell state by introducing polarization-dependent losses

stemming from detector dark counts (Supplementary Note 7). To quantify the influence of the converter on the final entangled state, it is useful to consider the case when only the detector part of the background is subtracted: we obtain $P = 95.1 \pm 1.3\%$, $F = 97.3 \pm 0.2\%$, and $F_{Bell} = 94.5 \pm 0.2\%$, which confirms that the conversion-induced noise has only a minor influence on the final state. If no background subtraction is applied, we get $P = 90.3 \pm 1.2\%$, $F = 94.8 \pm 0.2\%$, and $F_{Bell} = 92.2 \pm 0.2\%$.

Beyond the faithful QFC of ion–photon entanglement, the converter also renders possible the generation of maximally entangled states: we realize this by rotating the polarization of the 854 nm photons in a way that $|R\rangle$ and $|L\rangle$ correspond to the converter's interferometer arms. Then we reduce the conversion efficiency of the $|R\rangle$ arm by a factor of two to compensate the higher CGC, at the cost of one third of the photons. The resulting measured density matrix is displayed in Fig. 4b. The asymmetry in the diagonal elements disappeared, and we obtain $F_{Bell} = 98.2 \pm 0.2\%$, $P = 96.7 \pm 1.4\%$ (after subtraction of only detector dark counts: $F_{Bell} = 97.7 \pm 0.2\%$ and $P = 95.8 \pm 1.4\%$; without background subtraction: $F_{Bell} = 93.4 \pm 0.2\%$, $P = 87.8 \pm 1.3\%$). Thus, within the error bars, we have created a Bell state between ion and telecom photon with the same purity as the initially generated state, which proves that the converter leaves the ion–photon entanglement practically unaltered. Note that fidelity and purity in this measurement run are also in accordance with the process fidelity, which we attribute to a slightly more stable operation of the converter.

## Discussion

Our results demonstrate the operation of a complete quantum node that produces entangled states between a single trapped $Ca^+$ ion and a fiber-coupled telecom photon with a high fidelity. This constitutes a step towards the implementation of a fiber-based

repeater node consisting of two ions in remote traps. In future experiments, the entanglement generation rate might be enhanced with a cavity[5]. Furthermore, conversion to the telecom C-band at 1550 nm wavelength[30] shall be pursued for lower transmission losses enabling remote entanglement of ions over hundreds of kilometer fiber length. Moreover, spectral filters with narrower bandwidth combined with SSPDs with ultra-low dark-count rates[41] will lead to a much reduced background and higher SBR. Beyond these efforts, our techniques are transferable to a wide range of relevant platforms for quantum networks, such as other trapped-ion species ($Yb^+$, $Ba^+$), neutral atoms (Rb, Cs), color centers in diamond ($NV^-$, $SiV^-$), rare-earth ions in solids ($Pr^{3+}$, $Nd^{3+}$, $Eu^{3+}$), or quantum dots. Eventually, this approach opens the possibility to implement hybrid networks by coupling different quantum systems via a common bus wavelength in the telecom regime[42].

Recently, we became aware of a related experiment by Ikuta et al.[43] demonstrating entanglement between a cold atomic ensemble and a telecom photon via solid-state QFC.

## Methods

**Photon collection from the ion.** Single 854 nm photons emitted from the ion are collected by an in-vacuum high-numerical-aperture laser objective (HALO, Linos) with numerical aperture NA = 0.4 at a working distance of 13 mm from the ion, covering about 4% of the full solid angle. Collection efficiencies for photons emitted on the $\pi$ and $\sigma$ transitions are given by the dipole emission pattern. Orienting the quantization axis along the ion–HALO axis, the unnormalized polarization states of the emitted radiation are $\left|\psi_\pi^{(854)}\right\rangle = -\sin\theta\left|\hat{\theta}\right\rangle$ for $\Delta m = 0$ and $\left|\psi_{\sigma^\pm}^{(854)}\right\rangle = \frac{e^{\pm i\varphi}}{\sqrt{2}}\left(\cos\theta\left|\hat{\theta}\right\rangle \pm i\left|\hat{\varphi}\right\rangle\right)$ for $\Delta m = \pm 1$, where $\theta$ and $\varphi$ are the spherical polar and azimuthal angles of the direction of emission and $\left|\hat{\theta}\right\rangle$ and $\left|\hat{\varphi}\right\rangle$ are their respective spherical-coordinate unit vectors[4]. Thus, the collection of photons emitted on the $\pi$ transition is suppressed due to the single-mode fiber coupling, while the collection efficiency for the $|\Delta m| = 1$ transitions sums up to 6% with respect to spontaneous emission into full space. Taking into account the CGCs for $\sigma$- and $\pi$-decay, the

resulting collection probability for $\sigma$ emission is $0.6 \times 6\% = 3.6\%$. The single-mode–fiber coupling accounts for an additional factor of 39%, resulting in a total collection efficiency $\eta_{coll,tot.} \approx 1.4\%$

**OPO system at 2456 nm**. We employ a home-built continuous-wave optical parametric oscillator (OPO) delivering 1 W of single-mode, single-frequency output power at 2456 nm as pump source for the DFG process[44]. The OPO is pumped by a diode laser at 1081 nm (Toptica DL Pro) amplified with a Yb-doped fiber amplifier (LEA Photonics) with 15 W maximum output power. The OPO consists of a 40 mm long periodically poled LiNbO$_3$ crystal with 7 poling periods ($\Lambda = 31.7$ μm...32.7 μm) inside a signal-resonant bow-tie ring cavity. Tuning of the idler wavelength from 2310 to 2870 nm is achieved by changing the poling period, the crystal temperature or the cavity length via a piezo actuator. With this tuning range, we are able to cover the whole telecom O-band from 1260 to 1360 nm with the frequency converter. During the experiment, the OPO was operated at 2456 nm using a poling period of $\Lambda = 32.6$ μm at a temperature of 49 °C.

**Polarization-preserving frequency converter**. The input light is overlapped with a diagonally polarized stabilization laser at 854 nm in a fiber beam splitter with a transmission of 99% for the input and 1% for the stabilization. Behind the PBS we split the orthogonal polarization components, the H-polarization is rotated to the convertible V-polarization with a HWP. Both beams are coupled to the ridge waveguide via dichroic mirrors and aspheric zinc selenide (ZnSe) lenses with focal lengths of 11 mm and broadband anti-reflective (AR) coatings for all three wavelengths. The 40 mm long Zn:PPLN waveguide chip (NTT Electronics) with lateral dimensions of $9 \times 16$ μm consists of 12 ridge waveguides with 6 different poling periods $\Lambda = 22.60$ μm...22.85 μm (operating point: $\Lambda = 22.70$ μm, $T = 31$ °C) and AR-coatings for all wavelengths. The chip is temperature-stabilized and mounted on a 5-axis translation stage to achieve optimal mode-matching. The pump field at 2456 nm generated by the OPO is guided free-space to the converter. The beam passes a HWP and a rutile polarizer for power control. A 1600 nm long-pass filter used for clean-up and a telescope made of two AR-coated spherical CaF$_2$-lenses to achieve best possible coupling to the waveguide's fundamental spatial mode. The transmitted pump field is back-reflected by a mirror and recoupled to the waveguide to enable conversion in both directions. The converted light at 1310 nm is separated from the pump field with further dichroic mirrors, the former H-polarized light is back-rotated with a HWP, and both arms are superimposed with another PBS. A bulk 99:1 beam splitter separates a part of the light for the path-length stabilization. Variation of the path length causes a phase change between H- and V-polarization, which is measured with a QWP at 22.5°, a PBS and two photo diodes. From the two photo diode signals we calculate the contrast $\left(\frac{I_1 - I_2}{I_1 + I_2}\right)$, which serves as a power-independent error signal for the PID lock. The feedback on the path length is applied with a piezo actuator (PM) mounted beneath one of the mirrors. In the output arm a band pass filter (BPF, central wavelength: 1300 nm, bandwidth: 25 nm, Edmund optics) is followed by a chopper wheel, blocking alternatingly the stabilization laser and the photons, and another telescope to mode-match the beam to the fiber. Fiber coupling is realized with an AR-coated aspheric lens ($f = 8$ mm, Thorlabs). As a narrowband spectral filter a fiber Bragg grating (FBG, central wavelength tunable from 1307 to 1317 nm, linewidth: 25 GHz, Advanced optical solutions GmbH) is utilized. A drawing of the complete setup and the characterization of the converter as well as a detailed analysis of the efficiencies and losses is found in Supplementary Notes 2 and 3.

**Quantum-state tomography**. To perform quantum-state tomography, ion and photon are projected onto all 36 combinations of eigenstates of the Pauli operators $\sigma_x$, $\sigma_y$, and $\sigma_z$, where $\sigma_z$ represents the eigenbases ($|R\rangle/|L\rangle$ for the photon and $|-3/2\rangle/|1/2\rangle$ for the ion) and $\sigma_x/\sigma_y$ the superpositions of the latter. To compare the results with and without the frequency converter, we use two tomography setups for the projective measurement of the 854 nm and telecom photons (see Fig. 1). The tomography setup for 854 nm is inserted into the beam path via a flip mirror and consists of QWP, HWP, polarizer, and a silicon APD. The APD (SPCM-AQR-14, Perkin Elmer) has a quantum efficiency of $\eta_{APD} = 30\%$ and a dark-count rate of $\gamma_{DC,APD} = 117.7$ photons/s. The tomography setup for 1310 nm is realized with QWP, HWP, Wollaston prism and two commercial fiber-coupled superconducting-nanowire single-photon detectors (SSPD, Single Quantum). The quantum efficiencies and dark counts for SSPD1 (SSPD2) are 70(2)% (62(2)%) and 58.7 (56.4) photons/s, respectively. All waveplates are motorized and controlled via an Ethernet link to enable remote control of the complete experiment. The atomic state is analyzed with a combination of coherent pulses on the quadrupole transition at 729 nm and the RF-transition between the $S_{1/2}$-states followed by fluorescence detection (details see Supplementary Note 1). From these measurements, the density matrix is calculated via linear state reconstruction combined with a maximum-likelihood estimation (see Supplementary Note 6).

**Data availability**. All relevant data are available from the corresponding author on request.

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

## Acknowledgements

We thank Benjamin Kambs, Philipp Müller, and Jonas Becker for helpful discussions. This work was financially supported by the German Federal Ministry of Science and Education (Bundesministerium für Bildung und Forschung (BMBF)) within the project Q.com.Q (Contract No. 16KIS0127).

## Author contributions

M.B. and P.E. conducted the experiments and analyzed the data with help from S.K., M.B. and A.L. constructed the frequency converter; and P.E. and M.K. implemented the ion–photon entanglement sequence. S.K. developed a software toolbox for the state reconstruction. J.E. and C.B. conceived and supervised the project. M.B., P.E., J.E., and C.B. wrote the manuscript with input from all the authors.

## Additional information

**Competing interests:** The authors declare no competing interests.

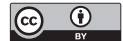

