## [Peer Review File · Nature Communications]

Reviewers' comments:

Reviewer #1 (Remarks to the Author):

Mathias Bock and colleagues report high-fidelity entanglement between a trapped ion and an emitted photon converted from its native 854 nm to 1310 nm via a polarization-entanglement preserving difference-frequency generating process.

Coherent light-matter interfaces at the single photon level and with telecom compatibility will form the core of future quantum networks, this work is therefore highly relevant. The authors have carried out a hero experiment with exceptional quality and efficiency in the state conversion. A number of similar experiments have been conducted but to my knowledge not for polarization entanglement and nowhere near this level of performance. I recommend publication with only some minor revision as detailed below.

(i) various fidelities and purities are reported in the main text after a detector background has been subtracted. Far from a smoking gun it turns out that this subtraction (as detailed in the supplement) makes only a minor difference to the results. Nevertheless, background subtraction is debatable because it's hardly ever implemented correctly (very often background is subtracted that is actually part of the signal). The details presented don't allow a judgement on that but I strongly suggest that both sets of values are reported in the main text rather than in the supplement.

(ii) Both the OPO crystal and the DFG crystal have a range of poling periods. I may be mistaken but I don't think the authors specify which poling period they used for their exact wavelength regime. From reading a previous publication from the same group I can't recall that the dispersion properties for the employed Zn:PPLN waveguide were referenced in detail, I would suggest to remedy that here.

(iii) I'm sure this isn't the only typo, but this one stood out because I hadn't encountered it before: it's "rutile", not "rutil" polarizer.

Reviewer #2 (Remarks to the Author):

The manuscript presents experimental results in which a trapped Ca ion is entangled with the polarization degree of freedom of an infrared photon, and that photon is subsequently converted to the telecom O-band (1322 nm) in a polarization insensitive manner. Quantum state tomography is used to demonstrate high-fidelity entanglement between the ion and the telecom photon. Similar entanglement of an atomic memory with polarization-preserving frequency conversion to telecom has previously been demonstrated via four-wave mixing (Ref. 13), but this is the first demonstration using solid-state quantum frequency conversion (along with the work reported in a similar manuscript, also submitted to Nature Communications). As such, it represents a significant step towards quantum networks that are capable of linking together quantum memories over long distances. I believe that the manuscript will be of interest not only to researchers working on quantum memories and frequency conversion but to the larger quantum information science community. This approach of using a Sagnac interferometer (also implemented by the other research team) may become a standard approach in other research groups.

The results are convincing (in particular, I find it impressive that the conversion process does not degrade the purity of the entangled state, within error bars) and in my opinion merit publication in Nature Communications. Below, I raise a few points about the manuscript that I imagine it will be straightforward for the authors to address.

The authors claim in their abstract that "[e]ntanglement between a stationary quantum system and a flying qubit is an essential ingredient of a quantum-repeater network" and in their introduction that "repeaters require efficient, low-noise quantum memories." I am sympathetic to the point that quantum repeaters without memories would be experimentally very challenging, but there have nevertheless been recent proposals (as well as earlier work by Knill and Laflamme) for memoryless and all-photonic quantum repeaters, so I don't think these statements are correct.

I understand that the manuscript length is limited, but it seems to me that the key features of the experimental setup belong in the main text. (The authors simply state: "The ion-photon interface is shown and explained in Fig. 1. Further details may be found in earlier publications and the method section.")

It is stated that the fidelities are calculated after subtraction of the detector dark counts. In my opinion, it needs to be clearly explained what is meant here (this is also not addressed in the supplemental information) since it's not possible to simply subtract dark counts from a fidelity. Do the authors mean that the density matrix in Fig. 2b is reconstructed not from their actual measurement results but from measurement results that have been modified to reflect the inferred effects of detector dark counts? If so, I would urge the authors to include their unmodified results in the main text along with their corrected results. Of course, it's reasonable to analyze the data so as to focus on the effects of the frequency conversion process. But quantum state tomography is understood to be a method which takes measured data as its starting point.

The authors compare their trapped ion system to that of Ref. 3, but perhaps more recent results from Monroe's group should also be referenced.

Reviewer #3 (Remarks to the Author):

The authors describe a method to prepare entanglement between a stationary qubit based on superposition of energy levels of a Calcium ion and the polarization of a single photon at telecommunication wavelengths, via quantum frequency conversion (QFC). A NIR photon initially entangled with the Ca ion is frequency converted to a telecom photon via a χ -2 process in a nonlinear waveguide. Mapping polarization into different propagation directions along the waveguide allows for polarization independent frequency conversion. Tomography of the joint state of the ion and telecom photon is used to infer ion-photon entanglement. The trapped ion functions as a long-lived quantum memory and the photon can be used to transmit quantum information over long distances.

The authors state that the entanglement of a stationary qubit with the polarization state of a telecom photon via QFC has remained an open challenge. They say that this is mainly due to the strong polarization dependence of the conversion process for efficient conversion. They also say that the integration of a QFC device into a quantum node had not been achieved. The authors conclude that their work is a major step towards the implementation of a fiber-based repeater node, and that their techniques can be transferred to other platforms for quantum networks, such as neutral atoms (Rb, Cs).

However, there is already a demonstration of a quantum memory entangled with a photon at telecommunication wavelength in polarization via QFC. The work reported in "Entanglement of Light-Shift Compensated Atomic Spin Waves with Telecom Light," by Dudin, et al., in *Phys. Rev. Lett.* 105, 260502 (2010), which is not discussed nor referenced in the present manuscript, describes the demonstration of a quantum memory based on an atomic Rb spin-wave qubit in an optical lattice entangled with a telecom photon. In the work by Dudin et al., efficient frequency conversion is achieved via four-wave mixing in Rb atoms, and entanglement is demonstrated via the violation of Bell's inequality after a long storage time of the stationary qubit, 10ms.

The work by Dudin, et al. already demonstrated the entanglement of a stationary qubit in a quantum memory with a telecom photon via QFC, which seems to be the main result of the present paper. In my opinion, this puts in question the impact of the current manuscript, and it is not clear that the

contribution of the paper beyond what has been reported by Dudin, et al. merits publication in Nature Communications.

Comments of Reviewer #1 -- NCOMMS-17-29329-T/Bock

Mathias Bock and colleagues report high-fidelity entanglement between a trapped ion and an emitted photon converted from its native 854 nm to 1310 nm via a polarization-entanglement preserving difference-frequency generating process.

Coherent light-matter interfaces at the single photon level and with telecom compatibility will form the core of future quantum networks, this work is therefore highly relevant. The authors have carried out a hero experiment with exceptional quality and efficiency in the state conversion. A number of similar experiments have been conducted but to my knowledge not for polarization entanglement and nowhere near this level of performance. I recommend publication with only some minor revision as detailed below.

We thank the referee for the strong support of our manuscript, the appreciation of the quality of our results and the recommendation for a publication in Nature Communications.

(i) various fidelities and purities are reported in the main text after a detector background has been subtracted. Far from a smoking gun it turns out that this subtraction (as detailed in the supplement) makes only a minor difference to the results. Nevertheless, background subtraction is debatable because it's hardly ever implemented correctly (very often background is subtracted that is actually part of the signal). The details presented don't allow a judgement on that but I strongly suggest that both sets of values are reported in the main text rather than in the supplement.

We thank both referees #1 and #2 for their comments on the background subtraction and answer them jointly here:

Both referees request the quotation of the uncorrected values in the main text. To account for this we added the following sentences to the manuscript:

1. p.3, paragraph on ion-photon entanglement without frequency conversion:

“Without background subtraction we deduce a purity $P = 92.1 \pm 1.6\%$ and fidelities $F = 95.9 \pm 0.3\%$ and $F_{\text{Bell}} = 93.3 \pm 0.3\%$. Even without background subtraction, all fidelities are many standard deviations above the classical threshold of 50%, as well as above the threshold of 70.7% necessary to violate Bell's inequalities. [...] A table summarizing all numbers along with a detailed account of their derivation is given in the supplementary information.”

2. p.4, paragraph on ion-photon entanglement with frequency conversion:

We stated that the numbers given in the original manuscript were derived after background subtraction. We further added:

“The background in these measurements has two contributions: a minor part of 6.5% due to conversion-induced noise and a major part of 93.5% stemming from detector dark counts (supplementary information). To quantify the influence of the converter

on the final entangled state, it is useful to consider the case when only the detector part of the background is subtracted: we obtain $P = 95.1 \pm 1.3\%$, $F = 97.3 \pm 0.2\%$ and $F_{\text{Bell}} = 94.5 \pm 0.2\%$, which confirms that the conversion-induced noise has only a minor influence on the final state. If no background subtraction is applied, we get $P = 90.3 \pm 1.2\%$, $F = 94.8 \pm 0.2\%$ and $F_{\text{Bell}} = 92.2 \pm 0.2\%$.”

3. p.4, paragraph on Bell state generation with frequency conversion:

“[...] (after subtraction of only detector dark counts: $F_{\text{Bell}} = 97.7 \pm 0.2\%$ and $P = 95.8 \pm 1.4\%$, without background subtraction: $F_{\text{Bell}} = 93.4 \pm 0.2\%$, $P = 87.8 \pm 1.3\%$)”

Moreover, we added a detailed description to the supplementary information (Sec. 7) on how the background subtraction was performed:

“As mentioned in the main text, background subtraction was applied to the raw data to demonstrate the functionality of our method. To perform this background subtraction, we reconstructed the density matrix as described in Sec. 6 from a modified raw data set. At first, we determine for each basis combination the average number of dark coincidences per time-bin. In Fig. S3a, for instance, this is done by integrating the number of coincidences outside the photon wavepacket (from $0\mu\text{s}$ to $2\mu\text{s}$) and dividing it by the number of time-bins. Now, we assume a Poissonian distribution for the dark coincidences (this is feasible as conversion-induced noise as well as detector dark counts occur independently) with the average number of dark coincidences as expectation value. With the distribution of the dark coincidences and the total number of coincidences, we calculate the distribution of the true signal coincidences, yielding expectation value and variance of the latter. From this point, we proceed as described in Sec. 6. This statistical approach is necessary on the one hand to get a proper error estimation and on the other hand to avoid negative values for the number of coincidences in some time-bins.

The following tables summarize again all fidelities and purities with and without the subtraction of the background (denoted in the tables as "Total BG subt." and "W/o BG subt.", respectively). In the measurements with the frequency converter, 93.5% of the background originates from the detectors and 6.5% from conversion-induced noise. Thus it is of interest to quantify solely the converter's influence on the state by subtracting only 93.5% of the background (in this case we multiply the average number of dark coincidences per bin with 0.935). This is denoted in the tables as "Detector DC subt.”

This statement clarifies our procedure for background subtraction which we believe is appropriate and scientifically correct.

(ii) both the OPO crystal and the DFG crystal have a range of poling periods. I may be mistaken but I don't think the authors specify which poling period they used for their exact wavelength regime. From reading a previous publication from the same group I can't recall that the dispersion properties for the employed Zn:PPLN waveguide were referenced in detail, I would suggest to remedy that here.

As requested we added the poling period of the OPO ($\Lambda = 32.6 \mu\text{m}$) and the DFG crystal ($\Lambda = 22.70 \mu\text{m}$) which we used in the experiment to the corresponding methods section.

(iii) I'm sure this isn't the only typo, but this one stood out because I hadn't encountered it before: it's "rutile", not "rutil" polarizer.

We corrected the above mentioned typo as well as further typos we recognized.

Comments of Reviewer #2 -- NCOMMS-17-29329-T/Bock

The manuscript presents experimental results in which a trapped Ca ion is entangled with the polarization degree of freedom of an infrared photon, and that photon is subsequently converted to the telecom O-band (1322 nm) in a polarization insensitive manner. Quantum state tomography is used to demonstrate high-fidelity entanglement between the ion and the telecom photon. Similar entanglement of an atomic memory with polarization-preserving frequency conversion to telecom has previously been demonstrated via four-wave mixing (Ref. 13), but this is the first demonstration using solid-state quantum frequency conversion (along with the work reported in a similar manuscript, also submitted to Nature Communications). As such, it represents a significant step towards quantum networks that are capable of linking together quantum memories over long distances. I believe that the manuscript will be of interest not only to researchers working on quantum memories and frequency conversion but to the larger quantum information science community. This approach of using a Sagnac interferometer (also implemented by the other research team) may become a standard approach in other research groups.

The results are convincing (in particular, I find it impressive that the conversion process does not degrade the purity of the entangled state, within error bars) and in my opinion merit publication in Nature Communications. Below, I raise a few points about the manuscript that I imagine it will be straightforward for the authors to address.

We thank referee #2 for pointing out the novelty, quality and impact of our results as well as for recommending publication in Nature Communications.

The authors claim in their abstract that "[e]ntanglement between a stationary quantum system and a flying qubit is an essential ingredient of a quantum-repeater network" and in their introduction that "repeaters require efficient, low-noise quantum memories." I am sympathetic to the point that quantum repeaters without memories would be experimentally very challenging, but there have nevertheless been recent proposals (as well as earlier work by Knill and Laflamme) for memoryless and all-photonic quantum repeaters, so I don't think these statements are correct.

We agree with the referee that our statement is too restrictive with respect to memoryless repeater schemes. To address this issue, we amended the main text (p.1, introduction) in the following way:

"While proposals for memoryless repeaters exist (e.g. Munro et al., Nat. Phot. 6, 777-781 (2012).), many currently pursued approaches require efficient, low-noise quantum memories as nodes that exchange quantum information via photonic channels (Sangouard et al., Rev. Mod. Phys. 83, 33 (2011))"

I understand that the manuscript length is limited, but it seems to me that the key features of the experimental setup belong in the main text. (The authors simply state: "The ion-photon interface is shown and explained in Fig. 1. Further details may be found in earlier publications and the method section.")

We agree with the referee that the main text lacks experimental details. Thus we revised the two paragraphs where the ion-photon interface (p.2) and the frequency converter (p.3) are described by adding experimental details mentioned only in the method section of the original manuscript. We like to point out, however, that we put an emphasis on providing details of the QFC device as this has not been described in earlier publications. The ion-photon interface, on the other hand, is extensively discussed and detailed in previous journal publications and we refer the reader to these references ([31], [32], [33] of the revised manuscript).

It is stated that the fidelities are calculated after subtraction of the detector dark counts. In my opinion, it needs to be clearly explained what is meant here (this is also not addressed in the supplemental information) since it's not possible to simply subtract dark counts from a fidelity. Do the authors mean that the density matrix in Fig. 2b is reconstructed not from their actual measurement results but from measurement results that have been modified to reflect the inferred effects of detector dark counts? If so, I would urge the authors to include their unmodified results in the main text along with their corrected results. Of course, it's reasonable to analyze the data so as to focus on the effects of the frequency conversion process. But quantum state tomography is understood to be a method which takes measured data as its starting point.

This comment is almost identical to the comment of referee #1 on background subtraction. We thus discussed both comments together in the answer to referee #1 above.

The authors compare their trapped ion system to that of Ref. 3, but perhaps more recent results from Monroe's group should also be referenced.

We agree with the referee that recent results by Chris Monroe's group, which features even higher entanglement rates, should be included. We changed the main text (p.3, paragraph on ion-photon entanglement) as follows:

"[...] which compares well with other Ca⁺-ion systems (Stute et al., Nature 485, 482-485 (2012)). One order of magnitude higher entanglement rates were reported in a Yb⁺-system, mainly due to a higher sequence repetition rate enabled by shorter cooling times and the use of ultrafast laser pulses (Hucul et al., Nat. Phys. 11, 37-42 (2015))."

Comments of Reviewer #3 -- NCOMMS-17-29329-T/Bock

The authors describe a method to prepare entanglement between a stationary qubit based on superposition of energy levels of a Calcium ion and the polarization of a single photon at telecommunication wavelengths, via quantum frequency conversion

(QFC). A NIR photon initially entangled with the Ca ion is frequency converted to a telecom photon via a χ^2 process in a nonlinear waveguide. Mapping polarization into different propagation directions along the waveguide allows for polarization independent frequency conversion. Tomography of the joint state of the ion and telecom photon is used to infer ion-photon entanglement. The trapped ion functions as a long-lived quantum memory and the photon can be used to transmit quantum information over long distances.

The authors state that the entanglement of a stationary qubit with the polarization state of a telecom photon via QFC has remained an open challenge. They say that this is mainly due to the strong polarization dependence of the conversion process for efficient conversion. They also say that the integration of a QFC device into a quantum node had not been achieved. The authors conclude that their work is a major step towards the implementation of a fiber-based repeater node, and that their techniques can be transferred to other platforms for quantum networks, such as neutral atoms (Rb, Cs).

However, there is already a demonstration of a quantum memory entangled with a photon at telecommunication wavelength in polarization via QFC. The work reported in "Entanglement of Light-Shift Compensated Atomic Spin Waves with Telecom Light" by Dudin, et al., in Phys. Rev. Lett. 105, 260502 (2010), which is not discussed nor referenced in the present manuscript, describes the demonstration of a quantum memory based on an atomic Rb spin-wave qubit in an optical lattice entangled with a telecom photon. In the work by Dudin et al., efficient frequency conversion is achieved via four-wave mixing in Rb atoms, and entanglement is demonstrated via the violation of Bell's inequality after a long storage time of the stationary qubit, 10ms.

The work by Dudin, et al. already demonstrated the entanglement of a stationary qubit in a quantum memory with a telecom photon via QFC, which seems to be the main result of the present paper. In my opinion, this puts in question the impact of the current manuscript, and it is not clear that the contribution of the paper beyond what has been reported by Dudin, et al. merits publication in Nature Communications.

Referee #3 claims that our manuscript does not merit publication in Nature Communications given the work by Dudin et al. which already demonstrated entanglement between a matter qubit and the polarization state of a telecom photon. We fully agree that this paper is an important step in this direction and thus added it to the references of our manuscript as well as to the discussion in the introduction of the main text.

Nevertheless, our opinion is that our approach based on solid-state QFC is well worth being published in Nature Communications as it provides a major advantage compared to the work by Dudin et al., namely the significantly better wavelength flexibility: the paper by Dudin et al. utilized a four-wave mixing process in a cold atomic Rubidium ensemble. The input wavelength in such an approach is fixed to a rather small interval around the transition wavelengths of neutral atoms (in particular the D-lines of Rubidium and Cesium). On the other hand, implementations of small quantum networks using trapped ions (Ca^+ , Yb^+ , Sr^+ ,...), NV centers in diamond or ensembles of rare earth ions as quantum nodes are heavily pushed forward by many groups worldwide. These systems have transition wavelengths, which usually differ from the alkali D-lines. Thus, the approach by Dudin et al. is not suitable for the

majority of candidate systems for quantum nodes. On the contrary, a solid-state QFC system does not require resonant optical transitions, but a transparent, highly nonlinear material with a highly developed manufacturing technology. State-of-the-art technology enables efficient frequency conversion to the telecom bands for input wavelengths between the blue and the near infrared spectral region and thus covers the relevant wavelength regime of all systems mentioned above. In summary, we believe that our approach is a more universal and flexible one, being highly relevant for future quantum networks, independently of which quantum system will establish itself as quantum node.

To recognize the important work of Dudin et al. and to point out the advantage of our approach, we modified the second and third paragraph in the introduction of the main text (p.1-2) as follows:

„The latter can be implemented either by four-wave mixing (FWM) using resonances in cold atomic ensembles [Radnaev et al., Nat. Phys. 6, 894 (2010), Dudin et al., Phys. Rev. Lett. 105, 260502 (2010).] or by a solid-state approach utilizing three-wave mixing in χ^2 - or four-wave mixing in χ^3 -nonlinear media [Citation Li et al., Nat. Phot. 10, 406-414 (2016).].”

“Using near-resonant QFC based on FWM in an atomic ensemble, entanglement of a spin wave qubit with the polarization state of a telecom photon has been realized [Dudin et al., Phys. Rev. Lett. 105, 260502 (2010).]. A corresponding implementation using solid-state QFC has remained an open challenge, despite being a highly desirable approach for its wavelength flexibility: while atomic ensembles are restricted to the particular transition wavelengths of neutral atoms, solid-state QFC can be adjusted to the system wavelength of other promising stationary quantum bits for quantum nodes, such as trapped ions, color centers in diamond or rare-earth ensembles. The main obstacle has been the strong polarization dependence of the χ^2 -process and the high demands on efficiency and noise properties of the converter. Despite successful attempts to overcome the polarization dependency [Citation Ramelow et al. Phys. Rev. A 85, 013845 (2012)., Albota et al. J. Opt. Soc. Am. B 23, 918-924 (2006)., Krutyanskiy et al. Appl. Phys. B 123, 228 (2017).], the integration of a solid-state QFC device that fulfills all above mentioned requirements into a quantum node has not been achieved.”

Furthermore we wish to point out an additional advantage of employing trapped ion systems for quantum nodes, i.e. their proven applicability for not only storing quantum states but also for processing of these states. We added the following phrase to the introduction:

“[...] importantly, single-ion qubits are directly addressable and thus allow quantum information processing via high-fidelity quantum gates [Citation Balance et al., Phys. Rev. Lett. 117, 060504 (2016)., Gaebler et al. Phys. Rev. Lett. 117, 060505 (2016).].”

Further comments and changes in the manuscript

1. We mentioned in the last sentence of the discussion, that our approach opens the possibility to implement hybrid networks by coupling different quantum systems via a common bus wavelength in the telecom regime. Recently, an experiment by the Riedmatten group (ICFO, Barcelona) connecting a Rubidium DLCZ memory with a Praseodymium AFC memory via two-step QFC has been published. We here added the corresponding reference ([43]).

2. During the revision process, we became aware of a related experiment by Ikuta et al. demonstrating entanglement between a cold atomic ensemble and a telecom photon via solid-state QFC. We added a note at the end of the main text along with the corresponding reference ([44]).

3. We became aware of the fact that the configuration of the frequency converter is not a true Sagnac configuration. In the literature the latter is not characterized by the use of a single nonlinear crystal with counter-propagating beams, but by an intrinsic phase stability, which requires a single ring interferometer loop for all three wavelengths (854nm, 1310nm and 2456nm). Thus we replaced the phrase “Sagnac configuration” by “single-crystal Mach-Zehnder configuration”, which is in our opinion a more suitable description of the converter’s configuration.

4. Throughout the manuscript we applied slight modifications to the wording in order to improve readability of the text.

REVIEWERS' COMMENTS:

Reviewer #1 (Remarks to the Author):

The authors have addressed my minor concerns, and I recommend publication of the article.

Let me add a final note on background subtraction though. I strongly disagree with the practice in general and don't see the necessity to go to all this trouble here in particular. The method the authors now describe in more detail does, in my experience, often overestimate the noise and thus make background-subtracted signal look better than it actually is.

Let's consider for a moment an ideal experiment, with zero noise, and zero loss, which produces coincident photon events of some form. Barring some details on whether such an experiment might be in a pulsed regime or not, this technique of measuring background at some time-shifted delay outside the window where coincident signal is expected would in such an experiment also return "noise", proportional to the random detection probability of uncorrelated signal in some time window. However, this "noise" isn't actually noise in our noise-less experiment: instead, it is signal measured at the wrong time. Subtracting this "noise" from the signal at $t=0$ would be completely nonsensical in this idealised example. So my claim would be that most experiments which estimate noise in this way do in fact overestimate noise, albeit not by much given the usually high loss rates and non-coincident signal contributions.

I acknowledge that this may not apply to the experiment at hand.

Reviewer #2 (Remarks to the Author):

After reviewing the revised manuscript, I am convinced that all of the points I raised have been thoroughly addressed, and I support publication of the manuscript.

Reviewer #3 (Remarks to the Author):

The authors revised the manuscript to include relevant work in quantum interfaces. The authors reviewed work including hybrid interfaces among different matter qubits through solid state frequency conversion. This revision puts their work in a broader context in long-distance quantum networks. The authors also discuss the differences with previous work and describe the advantages of their scheme for frequency conversion with high tunability over a large wavelength range compared to frequency conversion based on atomic systems. I recommend the revised manuscript for publication in Nature Communications.

The authors have addressed my minor concerns, and I recommend publication of the article.

Let me add a final note on background subtraction though. I strongly disagree with the practice in general and don't see the necessity to go to all this trouble here in particular. The method the authors now describe in more detail does, in my experience, often overestimate the noise and thus make background-subtracted signal look better than it actually is.

Let's consider for a moment an ideal experiment, with zero noise, and zero loss, which produces coincident photon events of some form. Barring some details on whether such an experiment might be in a pulsed regime or not, this technique of measuring background at some time-shifted delay outside the window where coincident signal is expected would in such an experiment also return "noise", proportional to the random detection probability of uncorrelated signal in some time window. However, this "noise" isn't actually noise in our noise-less experiment: instead, it is signal measured at the wrong time. Subtracting this "noise" from the signal at $t=0$ would be completely nonsensical in this idealised example. So my claim would be that most experiments which estimate noise in this way do in fact overestimate noise, albeit not by much given the usually high loss rates and non-coincident signal contributions.

I acknowledge that this may not apply to the experiment at hand.

We thank referee #1 for the renewed recommendation for publication in Nature Communications.

We are aware of the difficulty, as explained by the referee, of applying "correct" background subtraction in coincidence measurements in general. We also believe that this difficulty does (practically) not matter for our particular experiment, because it is carried out with an (approximate) triggered single-photon source.

Nevertheless, since it is an important and interesting scientific discussion, we would like to reply.

Let us first consider the specific example of a coincidence measurement (assuming the referee has this in mind), e.g. detecting photons on the two outputs of a continuous-wave SPDC entangled photon pair source. In this case, one would like to characterize (at least) two properties: the intrinsic purity (or Bell-state fidelity) of the pairs, and the device as a source of entanglement at a given generation rate. For the latter, the only background that may be subtracted is the one caused by the characterizing apparatus, i.e. by transmission loss (incl. detector inefficiency) and dark counts. For the former, subtracting the background that results from the "signal at the wrong time" is, in our view, correct. This background appears unavoidably outside the coincidence window, but it is also present inside this window: inasmuch as there are time-separated photons from different pairs, there are also coincident

photons from different pairs. (This statement must in fact be refined in order to account for the photon pair statistics, but we shall ignore this complication here.) This background leads to a decreasing signal-to-background ratio (SBR) as the generation rate increases, and thereby to a "perceived impurity". In order to find out the intrinsic impurity of the source, subtraction of this background contribution is therefore appropriate. The intrinsic impurity is an experimentally relevant quantity, as it may be obtained in the limit of a very low generation rate.

In our experiment, however, the situation is different: photons are generated from the ion within a specific time window of 3 μ s duration given by the excitation pulse on the S1/2 – P3/2 transition. The duration of the excitation is chosen such that there is a large probability for decay into the D5/2 state accompanied by the desired emission of a photon; at the same time the D5/2 state lifetime is so long (\sim 1s) that the probability for re-excitation and emission of a second photon is small. In this sense we generate triggered single photons with a timing uncertainty of 3 μ s. Background events outside this 3 μ s time window solely stem from detector dark counts (or additionally from conversion noise for the case of converted photons). In the experiment we indeed determined the background noise from intervals where the ion was prepared in the ground state and could not emit 854 nm photons. As this background is identical for all times subtraction of this contribution is appropriate to determine the intrinsic impurity of the source.

To clarify the issue pointed out by the referee, we added the following sentence to the Supplementary Note 7:

“This approach is justified in our experiment as we generate triggered single photons within a specified time window; thus, contrary to a photon-pair source, e.g. based on parametric conversion with random emission times, coincidences outside the photon wavepacket originate solely from detector dark-counts.”

Comments of Reviewer #2 -- NCOMMS-17-29329-T/Bock

After reviewing the revised manuscript, I am convinced that all of the points I raised have been thoroughly addressed, and I support publication of the manuscript.

We thank referee #2 for the renewed recommendation for publication in Nature Communications.

Comments of Reviewer #3 -- NCOMMS-17-29329-T/Bock

The authors revised the manuscript to include relevant work in quantum interfaces. The authors reviewed work including hybrid interfaces among different matter qubits through solid state frequency conversion. This revision puts their work in a broader context in long-distance quantum networks. The authors also discuss the differences with previous work and describe the advantages of their scheme for frequency conversion with high tunability over a large wavelength range compared to frequency

conversion based on atomic systems. I recommend the revised manuscript for publication in Nature Communications.

We thank referee #3 for his suggestions and the chance to improve our manuscript, as well as for supporting publication in Nature Communications.